# The Role of TGF-β3 in Radiation Response

**DOI:** 10.3390/ijms24087614

**Published:** 2023-04-20

**Authors:** Ingunn Hanson, Kathinka E. Pitman, Nina F. J. Edin

**Affiliations:** Department of Physics, University of Oslo, 0371 Oslo, Norway; ingunn.hanson@fys.uio.no (I.H.); k.e.pitman@fys.uio.no (K.E.P.)

**Keywords:** cell cycle regulation, chemotherapy, fibrosis, ionizing radiation, radiomitigation, transforming growth factor beta, TGF-β3

## Abstract

Transforming growth factor-beta 3 (TGF-β3) is a ubiquitously expressed multifunctional cytokine involved in a range of physiological and pathological conditions, including embryogenesis, cell cycle regulation, immunoregulation, and fibrogenesis. The cytotoxic effects of ionizing radiation are employed in cancer radiotherapy, but its actions also influence cellular signaling pathways, including that of TGF-β3. Furthermore, the cell cycle regulating and anti-fibrotic effects of TGF-β3 have identified it as a potential mitigator of radiation- and chemotherapy-induced toxicity in healthy tissue. This review discusses the radiobiology of TGF-β3, its induction in tissue by ionizing radiation, and its potential radioprotective and anti-fibrotic effects.

## 1. Introduction

The transforming growth factor-beta (TGF-β) family consists of multifunctional cytokines with potent effects on many cellular processes. More than 40 members have been identified, of which most are considered to belong to two major branches: TGF-β/Activin/Nodal/inhibins and BMP/GDFs (bone morphogenetic protein/growth and differentiation factors). Also included are left-right determination factor (Lefty) and anti-Müllerian hormone/Müllerian inhibiting substance (AMH/MIS) [1].

Three isoforms of TGF-β have been identified in humans. TGF-β3 was first identified at the cDNA level in 1988. The active domain of TGF-β3 shares 76% identity and 86% similarity with TGF-β1 and 79% identity and 91% similarity with TGF-β2 [2,3]. All three isoforms are ubiquitously expressed in human tissue; however, isoform-specific variations in tissue expression have been noted: TGF-β1 is most highly expressed in bone marrow and lymphoid tissue, TGF-β2 in brain and male reproductive tissue, and TGF-β3 in brain and female reproductive tissue [4]. 

Active TGF-βs produce functional effects influencing physiological and pathological processes, including cell cycle regulation, carcinogenesis, and immunoregulation. All TGF-β knockout mice fail to have viable litters, but the causes are distinct and not overlapping for the three isoforms. Knockout of TGF-β1 causes hematopoietic and vasculogenic defects and inflammation, while knockout of TGF-β2 causes multiple developmental defects including palatal shelf rotation [5,6]. TGF-β3 knockout mice die within 20 h after birth due to delayed pulmonary development and cleft palate with 100% penetration resulting in an inability to suckle [7,8]. Knock-in of TGF-β1 in the TGF-β3 expressing site only partially corrects the developmental defects displayed by TGF-β3 null mice, demonstrating the intrinsic functional differences between the two isoforms [9]. 

Radiation therapy is employed in the treatment of a wide range of cancers, either alone or in combination with other treatment modalities. Irradiation of cancerous as well as healthy tissue inevitably affects cellular signaling pathways, one of which is the TGF-β pathway. Understanding the effects of radiation is crucial to improving efficacy and reducing side effects from radiation therapy. The radiobiology of TGF-β1 was recently reviewed [10]; however, the isoform-specific effects of TGF-β3 were not covered. TGF-β3 has been observed to be both upregulated and activated by ionizing radiation. Furthermore, TGF-β3 affects the radiation response in cells and whole organisms. This review addresses the role of TGF-β3 in the response to ionizing radiation and the known underlying mechanisms.

## 2. Background

### 2.1. Induction, Upregulation, and Secretion

TGF-βs are upregulated at the mRNA level; however, the biochemical action of the transcribed protein is dependent on further processing and activation. All three TGF-β isoforms are synthesized as homodimeric pro-proteins (pro-TGF-β) that have a mass of 75 kDa and consist of TGF-β and the latency-associated protein (LAP). The LAP is cleaved from the mature TGF-β 24-kDa dimer in the trans Golgi by furin-type enzymes after which TGF-β remains non-covalently associated with LAP in an inactive state, also termed the small latent complex [11]. 

Latent TGF-β is mainly and most efficiently secreted chaperoned by latent TGF-β binding protein (LTBP) as part of the large latent complex (LLC) [12]. Secreted LLC is either soluble or sequestered in the extracellular matrix (ECM), and latent TGF-β is then stored extracellularly until it is activated [13]. Some latent TGF-β leaves the cell in the absence of LTBP, but at a much lower rate [14]. It is also unknown whether this TGF-β is functional [15]. 

More recently, TGF-β1 has been found in some cases to be secreted in extracellular vesicles (EVs), reviewed in [16]. In one study, EVs from healthy fibroblasts were found to contribute 53.4% to 86.3% of the total TGF-β1 present in the cancer cell supernatant [17]. The TGF-β1 in the EVs was shown to be associated with a transmembrane protein. In order to bind to TGF-β1 receptors at the cellular surface, TGF-β1 was released from the EVs mediated by digestion of the transmembrane protein by proteinase K [17]. Another study showed that TGF-β1 was associated with membranes in EVs from mast cells by surface glycans. Under acidic conditions, it was activated from the latent form and dissociated from the EVs. Interestingly, the exosome-associated TGF-β1 was found to be more potent than the free form of the molecule [18].

TGF-β3 has been detected in EVs from human pulmonary artery smooth muscle cells, head and neck squamous cell carcinoma (HNSCC) cell lines, HNSCC patient plasma, and T-47D breast cancer cells [19,20,21].

### 2.2. Latency and Activation

Activation of latent TGF-β takes place when the TGF-β dimer is released from LAP. This activation can occur as a consequence of high or low pH, high temperature, or ionizing radiation. It can be mediated by biological molecules, including integrins, proteases, thrombospondin-1, and reactive oxygen species (ROS) [11,22,23]. The activation mechanisms for TGF-β1 have been extensively studied, but whether the activation mechanisms for TGF-β2 and TGF-β3 are identical is not clear. 

Jobling et al. suggested that activation of TGF-β by ionizing radiation is due to the generation of ROS and found that ROS activate latent TGF-β1, but not latent TGF-β2 or TGF-β3 [24]. 

The binding of integrins αvβ6 or αvβ8 to the RGD motif of LAP is an established mechanism of TGF-β1 activation [11]. The TGF-β1 LAP also binds integrins αvβ1, αvβ3, αvβ5, and α8β1 [25]. TGF-β3 LAP also has the RGD motif and has been seen to bind to integrins αvβ6 or αvβ8 [26]. Experiments with integrin-deficient mice demonstrated a role for αvβ6 and αvβ8 integrin activation of both TGF-β1 and TGF-β3 in maintaining immune homeostasis and organ development [27]. The LAP of TGF-β2 has an R-S substitution at the RGD site and TGF-β2 does not appear to be activated by integrins [28].

In addition to integrins, several proteins belonging to the metzincin superfamily have been demonstrated to activate latent TGF-βs. Matrix metalloproteinase (MMP) 2, MMP3, MMP9, MMP14, and ADAMTS1 (a disintegrin and metalloproteinase with thrombospondin motifs 1) have been observed to activate TGF-β1 [29,30,31,32,33,34,35], while MMP9 and MMP13 have been observed to activate TGF-β2 [36,37]. As of today, MMP9 is the only metzincin family member demonstrated to activate TGF-β3 [37]. In addition, ADAM17 has been shown to negatively impact the concentration of the TGF-β receptor ALK5 [38,39,40], thereby regulating TGF-β signaling.

While the receptor-binding domains of the TGF-βs are highly homologous, the respective LAP domains have only 42% to 56% amino acid similarity [11,41]. It is, therefore, more likely that the LAPs rather than the receptor-binding domain are responsible for the isoform-specific functions. In corroboration, it was shown that while the LAP of TGF-β1 could inhibit all three isoforms, the LAPs of TGF-β2 and TGF-β3 had a much less inhibitory effect even on their respective TGF-β [41].

### 2.3. Receptor Binding and Signaling

When activated, the TGF-β family ligands first bind to a type II receptor (TβRII), which recruits and phosphorylates a type I receptor (TβRI) in a heteromeric complex of the two transmembrane serine/threonine kinase receptors [42]. There are seven type I receptors (activin receptor-like kinases ALK1–ALK7) and five type II receptors (ActRIIA, ActRIIB, BMPRII, TGFβRII, and AMHRII). Generally, ALK4, ALK5, and ALK7 bind to TGF-β, nodal, and activin ligands, whereas ALK1, ALK2, ALK3, and ALK6 bind BMPs and GDFs [43]. The signal is then mainly propagated through the phosphorylation of intracellular Smad proteins [44]. 

Despite the structural similarities with TGF-β2, TGF-β3 resembles TGF-β1 in the way it is capable of binding to TβRII directly before recruiting TβRI, while TGF-β2 requires a non-signaling co-receptor TβRIII (beta glycan) to facilitate binding to TβRII/TβRI [28]. The binding to TβRII was found to be nearly identical for TGF-β3 and TGF-β1, but relative to the canonical closed conformation for TGF-β1, TβRII-bound TGF-β3 shows an altered arrangement of its monomeric subunits, designated the “open” conformation [45]. It has been suggested that the difference in the structure of the TGF-β3/TβRII complex may engage downstream signaling pathways differently than the TGF-β1/TβRII complex and thus lead to quantitatively and qualitatively different biological responses [46].

In most cell types, the TGFβRII-bound TGF-β1 or TGF-β3 signals via binding to ALK5, followed by Smad 2/3 signaling. A small difference between TGF-β1 and TGF-β3 in the dissociation rate constant for the recruitment of ALK5 has been observed [47]. 

ALK1 was first identified as a receptor for BMP-9 and BMP-10 and was shown to mediate downstream signals associated with Smad 1/5/8 [48]. Although less frequently reported than ALK5 binding, both TGF-β1 and TGF-β3 have been identified as ligands for ALK1 [21,49,50,51]. For endothelial cells, ALK1 signaling has been seen to result in increased cell proliferation and migration, while ALK5 signaling inhibited proliferation and migration through Smad 2/3 signaling. Endothelial cells lacking ALK5 were found to be deficient in TGF-β1/ALK1-induced responses, suggesting that ALK1 directly antagonizes ALK5 signaling and that ALK5 is required for optimal ALK1 activation by TGF-β1. ALK1 has since been shown to play a similar role in chondrocytes [52].

While binding of TGFβRII usually precedes binding of type I receptors ALK1 or ALK5, several studies have reported functional effects of TGF-β/ALK1 signaling that were not affected by deletion or inhibition of TGFβRII [21,53]. ALK1 could potentially be acting with an unidentified type II receptor; however, TGFβRII does not have any close relatives in humans [43].

In addition to the so-called “canonical” TGF-β signaling pathways described above, which are characterized by phosphorylation of Smads, TGF-βs can signal through several other, “non-canonical” pathways. These include the extracellular signal-regulated (Erk)/mitogen-activated protein kinase (MAPK) pathway, the c-Jun N-terminal kinase (JNK) and p38 MAPK signaling cascade, Rho-like GTPase signaling pathways and the phosphatidylinositol-3-kinase/Akt pathway [54]. 

## 3. Functional Effects of TGF-β3

### 3.1. Immunoregulation and Inflammation

TGF-βs are expressed by immune cells and are thought to play an important immunoregulatory role. TGF-β1 is the most extensively described isoform in regard to immunity, mainly because TGF-β1 null mice develop a fatal systemic autoimmune disease [5]. While the role of TGF-β2 is currently considered to be insignificant in the regulation of the immune system because of its negligible expression in immune cells, TGF-β3 is emerging as a potentially important immunoregulator with functions that differ from those of TGF-β1 in many contexts [55]. 

Several cells of the immune system have been reported to produce TGF-β3 either at the mRNA or protein level, and the role of TGF-β3 has been reported to be both pro- and anti-inflammatory. Interleukin-17-producing helper T cells (Th17 cells) are generated from naïve CD4+ T cells via several cytokines, including TGF-β. Lee et al. found that while generation with TGF-β1 produced a type of Th17 cell that did not readily induce autoimmune disease, TGF-β3 generated a functionally distinct type of Th17 cell that was highly pathogenic [56]. In B cells, TGF-β3 is thought to enhance antibody production under certain circumstances and can possess bifunctional effects depending on concentration [57]. 

The anti-inflammatory role of TGF-β3 has been highlighted through its ability to inhibit differentiation of forkhead box P3-expressing CD4^+^ T cells [58], its potential to inhibit B cell proliferation and antibody production [59], and its role in lymphocyte-activation gene 3^+^ regulatory T cells (Treg) mediated immune suppression [60]. TGF-β3 was also observed to suppress the development of germinal center B cells and thereby inhibiting the T cell-dependent immune response, a function which was not associated with TGF-β1 [57]. 

### 3.2. Fibrosis and Wound Healing

Radiation fibrosis is a late complication of radiotherapy, with potentially debilitating effects. While it is widely accepted that the action of TGF-β1 is pro-fibrotic, and TGF-β2 is generally believed to aid in this action, conflicting reports exist regarding TGF-β3 [61,62,63,64]. Several studies have reported pro-fibrotic effects of TGF-β3: Sun et al. registered upregulation of TGF-β2 and TGF-β3, but not TGF-β1, in human idiopathic pulmonary fibrosis and non-alcoholic fatty liver disease. In addition, they demonstrated pro-fibrotic effects of TGF-β3 in mouse models of these fibrotic diseases [41]. Guo et al. proposed that TGF-β3 promoted liver fibrosis by downregulation of MMP13 [65]. Reggio et al. reported an increased expression of extracellular matrix and inflammatory genes after treating cultured adipocytes and endothelial cells with TGF-β3 [66]. 

Others, however, have argued the regulatory or anti-fibrotic roles of TGF-β3 [67,68,69,70]. Although they detected the increased expression of TGF-β3 in tissue samples from human myocardial infarction patients, Xue et al. found that TGF-β3 suppressed cardiac fibroblasts and decreased fibrotic markers in an in vitro model [71]. Wu et al. concluded that transplanted mesenchymal stem cells reduced skin fibrosis through the action of TGF-β3 [72]. Escansany et al. observed renal fibrosis in TGF-β3 deficient mice, in addition to impaired lipid metabolism [73]. 

Wound healing and scar formation are closely tied to the fibrotic response. Early studies indicated that scarless healing observed in mammalian embryos was due to elevated levels of TGF-β3 in fetal tissue [74,75]. Furthermore, Shah et al. demonstrated that cutaneous scarring in rats was reduced by the addition of exogenous TGF-β3 or by the inhibition of TGF-β1 or -β2 [76]. Indeed, recombinant human TGF-β3 under the trade name Avotermin was developed as an anti-scarring agent, showing potential to provide an accelerated and permanent improvement in scarring in several phase I/II trials [77,78,79], but failed the phase III trial [80]. Despite the clinical failure of Avotermin, several more recent studies have supported TGF-β3′s anti-scarring effects [72,81,82,83,84].

### 3.3. Growth Regulation and Carcinogenesis

Shortly after its discovery, TGF-β1 was identified as a growth stimulator in fibroblasts, but a potent growth inhibitor in other cell types, including epithelial, lymphoid, and endothelial cells [85]. Early studies indicated identical functions for TGF-β2 and TGF-β3, but later research has elucidated isoform-specific variations in the growth-regulatory effects of TGF-βs. The addition of exogenous TGF-β3, but not TGF-β1, increased chondrocyte proliferation in neonatal murine cartilage in vitro [86]. Similarly, TGF-β3 increased chondrocyte proliferation in murine cranial suture-derived mesenchymal cells in vitro, while TGF-β1 had the opposite effect [87]. In contrast, mesenchymal stem cells were found to inhibit keloid fibroblast proliferation in vitro through a TGF-β3-dependent mechanism [72], and the addition of exogenous TGF-β3 reduced the proliferation of cultured taste bud epithelial cells [88]. Treatment with TGF-β3 also inhibited the proliferation of smooth muscle cells in vitro [89,90].

When considering the role of TGF-β3 in radiation cancer therapy, it is also of interest to consider its role in the disease itself. Research regarding the role of TGF-β3 in carcinogenesis and cancer progression is in many cases obscured by an assumption that the three TGF-β isoforms have identical functions and are interchangeable. This assumption is not necessarily true and can often be misleading. Furthermore, the majority of data referenced regarding the tumorigenic role of TGF-β3 originates from protein or mRNA analyses of tumor biopsies and are thus correlative [46]. Considerable caution should be applied before interpreting the data as causative. 

In an animal model of cutaneous melanoma, TGF-β3, unlike TGF-β1 and TGF-β2, was induced in cells adjacent to tumor tissue by factors released by the malignant cells, suggesting a responsive rather than a causative role of TGF-β3 [91]. Another study observed decreased TGF-β3 expression in the skin overlying malignant melanoma and suggested that melanoma cells either suppressed TGF-β3 expression or that the lack of TGF-β3 expression itself promoted melanocyte proliferation [92]. 

In several studies identifying genes associated with prognosis in breast cancer patients, increased TGF-β3 mRNA expression was correlated with a longer interval to distal metastases [93,94,95]. However, increased TGF-β3 protein levels have been observed to correlate with the incidence of lymph node metastases [96], a decrease in survival time [97], and overall survival [98] in breast cancer patients. One study found that the increased expression of TGF-β3 mRNA could lead to ovarian carcinogenesis and tumor progression [99], while another concluded that the increased TGF-β3 expression induced by progestin treatment was associated with a lower risk of developing ovarian cancer in macaques [100]. An analysis of osteosarcoma biopsies revealed a correlation between TGF-β3 protein content and lung metastasis incidence [101]. Leiomyoma samples have been shown to have a higher expression of TGF-β3 mRNA and protein than normal myometrium, and the addition of TGF-β3 increased the proliferation of cultured leiomyoma cells [90,102,103]. 

## 4. TGF-β3 and Radiation

### 4.1. Changes in TGF-β3 Expression after Ionizing Radiation

While ionizing radiation is known to be both cytotoxic and mutagenic, it also activates a large variety of transcriptional pathways in exposed cells and tissues, including TGF-β pathways. Table 1 contains an overview of studies where the endogenous expression of TGF-β3 after low linear energy transfer (LET) irradiation (250 kV X-rays, ^137^Cs, or ^60^Co) was investigated in vitro. In Table 2, in vivo studies investigating the expression of TGF-β3 after low LET irradiation (250 kV X-rays, 6 MeV x-rays, ^137^Cs, or ^60^Co) are summarized. 

One group investigated the expression of different TGF-β isoforms after the irradiation of rat kidney cells and its effect on radiation-induced nephropathy [104,105,106]. In mesangial cells, they found that mRNA levels for TGF-β3 gradually decreased until 48 h after irradiation, regardless of dose (5–20 Gy). In contrast, TGF-β1 mRNA gradually increased within the same time frame, also in a dose-independent manner. Simultaneously, ECM-associated fibronectin and biglycan mRNA were upregulated. Later, they found a similar pattern of TGF-β1 and TGF-β3 gene expression, associated with an increase in collagen I, plasminogen activator inhibitor 1, and MMP2 mRNA in tubule epithelial cells. However, when investigating the level of TGF-β protein in similar scenarios, they found a modest dose-dependent increase in total protein amounts, with no change in the amount of active TGF-β. Interestingly, the majority of active TGF-β protein belonged to the TGF-β3 isoform, both in control and irradiated cells. 

More recently, Thao et al. found an increase in TGF-β3 mRNA after an irradiation dose of 8 Gy in mouse glioma cell line GL261 [107]. The same response was seen for TGF-β2, ALK5, TGFβRII, Smad 3, and Smad 4 mRNA. Smad 7, an inhibitor of TGF-β/Smad signaling, was downregulated. They also showed that the addition of exogenous TGF-β2 induced the expression of several genes related to DNA repair, with an accompanying increase in protein levels. By inhibiting ALK5 and TGFβRII using a dual inhibitor, they suppressed the DNA repair gene expression and, when combined with radiation, increased tumor control in a xenograft mouse model, compared to either treatment alone. With this, they point to the potential therapeutic use of inhibiting the canonical TGF-β pathway in cancer radiotherapy. 

In breast cancer cell lines MCF-7 and MDA-MB-231, Yadav et al. demonstrated an increase in TGF-β3 gene expression with an accompanying increase in TGF-β3 supernatant protein, seven days after an irradiation dose of 6 Gy [108]. They also observed an upregulation of TGF-β2, ALK5, and TGFβRII mRNA, and a significant, albeit much more modest, increase in TGF-β1 and TGF-β2 supernatant protein content for both cell lines. These changes were accompanied by increases in proliferation and migration as well as in cell death by apoptosis or necrosis. The increase in proliferation could be inhibited by the inhibition of ALK5, confirming the involvement of the TGF-β pathway. When further challenged by a new dose of 6 Gy, the cells showed increased radioresistance compared to the first dose. 

Finkelstein, Rubin, and Johnston et al. were the first to demonstrate time-dependent changes in TGF-β3 mRNA after irradiation in vivo in C57BL/6 mouse lung [109,110,111]. In one study, they found a decrease in gene expression of TGF-β3 as well as TGF-β1 and TGF-β2 only one day after radiation doses of 5 and 12.5 Gy. For TGF-β3, this changed to an increase 14 days after both doses, while an increase was only evident after the higher dose for TGF-β2, and TGF-β1 levels had returned to baseline. After 14 days, they found an accompanying increase in gene expression for collagen I, III, and IV (CI, CIII, CIV), as well as fibronectin (FN), for one or both doses. No changes in tissue morphology or collagen deposition were evident at this stage, and neither was active inflammation. When expanding the study to include a period of 26 weeks after irradiation with 12.5 Gy, they found that the TGF-β3 mRNA level was maintained at an elevated level for 8 weeks and that TGF-β1 was also elevated at this time point. Within 16 weeks post irradiation, mRNA levels of both isoforms had returned to baseline levels. Moreover, CI, CIII, CIV, and FN mRNA were at baseline levels after 16 weeks but were upregulated again after 26 weeks. For the 5 Gy dose, the response was more modest, with only TGF-β1 increased after 8 weeks, and only CI after 26 weeks. 

Later, the same group investigated the differences between the radiosensitive strand C57BL/6 and the more radioresistant C3H/HeJ mice. While the changes in mRNA for TGF-β1, TGF-β3, CI, CIII, CIV, and FN were confirmed in C57BL/6 mice, C3H/HeJ mice displayed no significant changes in the TGF-β levels of either isoform. With regard to the ECM protein mRNA expression, C3H/HeJ mice displayed a slight reduction in CI and CIII 26 weeks after 12.5 Gy following an increase in CIII after 8 weeks and an increase in FN after 16 weeks. 

Several studies have been conducted regarding the expression of TGF-β3 in gastrointestinal (GI) tissue after irradiation in vivo. Ruifrok et al. studied the relationship between total growth factor content and proliferation in mouse jejunum crypts and villi [112]. Immediately after a 5 Gy irradiation dose, they observed a decrease in the mitotic index accompanied by an increase in the apoptotic index in the irradiated crypts, while the crypt cellularity first declined before it increased to 150% of baseline 5 days after irradiation. In the villi, no mitotic or apoptotic activities were observed regardless of irradiation, but the cellularity declined three days after treatment, before recovering within seven days. Levels of TGF-β1, -β2, and -β3 were higher in villi than in crypts, with TGF-β3 levels as the highest. The levels of each TGF-β varied over the first eight days after irradiation, with a decline on day 1, then a gradual recovery before a new decline on day 6. The pattern was strongest for TGF-β3. In the crypts, but not the villi, levels of ALK5 and TGFβRII increased one to two days after irradiation, before gradually declining to baseline within day 6. Interestingly, TGF-β3 levels in the villi were negatively correlated with crypt cellularity. Based on the observed pattern of growth factor content and crypt cell proliferation, they argue that decreased TGF-β production may trigger a proliferative response in the following days. 

Wang et al. studied changes in TGF-β mRNA and protein expression 2 and 26 weeks after irradiating a portion of mouse intestine with 12 and 21 Gy [113]. They found an increase in mRNA for all three isoforms two weeks after irradiation, but the increase was weaker for TGF-β2 and TGF-β3 levels. While these returned to baseline within 26 weeks, TGF-β1 expression was maintained at an elevated level. They observed particularly high expression of TGF-β1, and to a lesser degree TGF-β2 and TGF-β3, in cells adjacent to radiation-induced ulcers. When measuring TGF-β proteins, TGF-β1 levels were substantially increased 2 and 26 weeks after irradiation, whereas TGF-β2 and TGF-β3 showed minimal changes. 

Okunieff et al. found increases in mRNA for all TGF-β isoforms 18–25 weeks after 12.5 and 13.5 Gy irradiation of mouse bowel [114]. The increase in growth factor expression was accompanied by tissue toxicity evident by histopathology. 

When studying partially irradiated rat liver, Seong et al. observed different patterns of mRNA upregulation for TGF-β1 and TGF-β3 [115]. While TGF-β3 mRNA levels peaked at 4.8-fold 7 days after 25 Gy irradiation before declining to a near-baseline level within 15 days, TGF-β1 mRNA levels steadily increased throughout the measuring period and peaked at 3.6-fold at the last measuring point at 28 days. Staining for TGF-β protein increased around the central vein on day 7 for TGF-β3, and on day 28 for TGF-β1. No change in tissue morphology, collagen deposition, or active inflammation was observed in this study. 

Schultze-Mosgau et al. investigated TGF-β3 protein content during wound healing in preoperatively irradiated rat skin [116]. As expected, fractionated 30 Gy irradiation 4 weeks before surgery impaired the wound healing function of the skin. In the graft bed, but not in the transition between grafted skin and graft bed, TGF-β3 levels were lower in preoperatively irradiated than in non-irradiated tissue from day 7 to day 28 after surgery. In skin that had been irradiated, but not grafted, TGF-β3 levels were observed to be lower than controls from immediately after irradiation and until day 7 before returning to baseline. 

In a study of the protective effects of Ginkgo biloba extract, Yirmibesoglu observed an increase in TGF-β3 in rat skin after a 36 Gy radiation dose [117]. This study group, which was used as a radiation control and not treated with Ginkgo biloba extract, also displayed higher dermatitis scores than the sham-irradiated group.

In summary, in vitro studies reported that low LET irradiation was found to increase TGF-β3 mRNA levels in human MCF-7 and MDA-MB-231 cells and mouse GL261 cells, while they were decreased in rat mesangial cells and rat NRK52E cells. TGF-β3 protein levels were found to be increased in human MCF-7 and MDA-MB-231 cells after irradiation, while total and active levels of TGF-β3 protein remained relatively constant in rat mesangial cells. In vivo, for the GI, TGF-β3 mRNA levels were seen to increase in Sprague-Dawley rats and in DBA mice after irradiation. For TGF-β3 protein levels, a decrease was seen in intestinal villi, but an increase in intestinal crypts in C3Hf/Kam mice. For lung tissue, TGF-β3 mRNA levels were found to be elevated in C57BL/6 mice after irradiation, but not in radioresistant C3H/HeJ mice. For the liver, a similar time-dependent increase in TGF-β3 mRNA levels was seen in Sprague-Dawley rats. For skin in contrast, a decrease in TGF-β3 mRNA levels was observed in Wistar rats after fractionated irradiation. However, an increase in TGF-β3 protein level was found in the skin of the same rat model after a single dose.

**Table 1 ijms-24-07614-t001:** Studies of endogenous transforming growth factor-beta 3 (TGF-β3) expression after irradiation in vitro. ALK: activin-like kinase.

Cell Line	Radiation Regime	Results	Reference
MCF-7 and MDA-MB-231	2–10 Gy, ^60^Co	TGF-β3 mRNA increased twofold in MCF-7 cells and tenfold in MDA-MB-231 cells 7 days after 6 Gy irradiation. Similar increases were found in protein levels in the supernatant. Inhibition of ALK5 counteracted the increase in the proliferation of surviving cells induced by 6 Gy irradiation.	[108]
T98G and T-47D	0.2 Gy, ^60^Co	Protracted low dose-rate irradiation increased the amount of active TGF-β3 in the cytoplasm of T98G and T-47D cells.	[118]
Rat Mesangial Cells	5–20 Gy, ^137^Cs	TGF-β3 mRNA decreased in a dose-independent manner 24 h after irradiation, with a continued reduction after 48 h.	[104]
Rat Mesangial Cells	0.5–20 Gy, ^137^Cs	TGF-β3 mRNA was downregulated in a dose-dependent manner 24 h after irradiation. However, the level of active TGF-β protein remained relatively constant, and the majority of the active protein was of the TGF-β3 isoform.	[105]
NRK52E	1–10 Gy, ^137^Cs	In quiescent rat tubule epithelial cells, TGF-β3 mRNA levels were downregulated 3, 7 and 24 h after 10 Gy irradiation, and reached baseline after 48 h.	[106]
GL261	8 Gy, 200 kV X-rays	TGF-β3 mRNA was upregulated after 8 Gy.	[107]

**Table 2 ijms-24-07614-t002:** Studies of endogenous TGF-β3 expression after irradiation in vivo. GI: Gastro-intestinal.

Organ System	Model	Radiation Regime	Results	Reference
GI	C3Hf/Kam mice	5 Gy, 250 kV X-rays	TGF-β3 protein content decreased 6 days after irradiation in intestinal villi. In intestinal crypts, TGF-β3 protein content increased 1–2 days after irradiation. A negative correlation between TGF-β3 in the crypts and crypt cellularity was observed.	[112]
GI	Sprague-Dawley rats	12–21 Gy, 250 kV X-rays	TGF-β3 mRNA expression was increased in rat intestines two weeks after irradiation.	[113]
GI	DBA mice	12.5–13.5 Gy, ^60^Co	TGF-β3 mRNA significantly upregulated in the bowel 18–25 weeks after 12.5 Gy.	[114]
Lung	C57BL/6 mice	5–12.5 Gy, ^137^Cs	Evidence of a dose-dependent reduction in TGF-β3 mRNA in mouse lung 1 day post irradiation. After 14 days, TGF-β3 mRNA was elevated tenfold after 5 Gy, and fourfold after 12.5 Gy.	[109]
Lung	C57BL/6 mice	5–12.5 Gy, ^137^Cs	TGF-β3 mRNA was elevated in mouse lung 1 week after 12.5 Gy irradiation and persisted at 8 weeks before returning to baseline 16 weeks post irradiation. After 5 Gy, TGF-β3 mRNA increased tenfold after 2 weeks, before returning to baseline at 8 weeks post irradiation.	[110]
Lung	C57BL/6 and C3H/HeJ mice	5–12.5 Gy, ^137^Cs	TGF-β3 mRNA increased twofold 8 weeks after 12.5 Gy in the lung of radiosensitive C57BL/6 mice and returned to baseline within 16 weeks. After 5 Gy, TGF-β3 mRNA was significantly reduced in C57BL/6 mice after 16 weeks and returned to baseline within 26 weeks. Radioresistant C3H/HeJ mice displayed no significant change in lung TGF-β levels.	[111]
Liver	Sprague-Dawley rats	25 Gy, ^60^Co	TGF-β3 mRNA increased steadily from 1 day after irradiation, peaked at 4.8-fold on day 7, before gradually declining until day 28. On day 7, increased positive staining for TGF-β3 protein was observed.	[115]
Skin	Wistar rats	3 × 10 Gy (=30 Gy) over 14 days, 6 MeV protons	TGF-β3 protein content was reduced in rat skin starting directly after irradiation and up to 11 days. In preoperatively irradiated skin, TGF-β3 levels were still reduced 7–28 days after surgery (35–56 days after irradiation).	[116]
Skin	Wistar rats	36 Gy, ^60^Co	TGF-β3 protein content in rat skin increased 21 days after irradiation.	[117]

### 4.2. Treatment with Exogenous TGF-β3 and Irradiation 

Due to its apparent role in biological radiation response, TGF-β3 has been identified as a potential target for modification of the radiation sensitivity of cells and tissue, mainly through the addition of recombinant TGF-β3. Table 3 displays in vitro studies where the effect of the addition of exogenous TGF-β3 in combination with low LET (^137^Cs or ^60^Co) radiation was investigated. 

Robson et al. observed differing responses to treatment with exogenous TGF-β3 between normal and malignant colorectal cells [119]. While incubation for 24 h with 5 ng/mL TGF-β3 decreased the plating efficiency of normal rat IEC6 cells, it was increased for human colonic carcinoma Widr cells. In Widr cells, TGF-β3 treatment led to an increase in radiosensitivity as apparent by a reduction in the shoulder region of the survival curve. No effect was seen on the radiation response of IEC6 cells. Increasing the exposure time to TGF-β3 led to an increased radiosensitivity at the 5 Gy dose point for Widr cells, but not IEC6 cells. In IEC6 cells, TGF-β3 treatment resulted in an accumulation of 96% of the cells in the G1 phase after 5 Gy irradiation, while Widr cells displayed a 2–3 fold increase in cells in the S phase under identical conditions. Treatment with TGF-β1 did not affect the radiosensitivity or the cell cycle distribution after 5 Gy of either cell line. 

Table 4 summarizes in vivo studies where exogenous TGF-β3 was utilized in combination with low LET radiation (300 kV X-rays, ^137^Cs, or ^60^Co).

Potten, Booth, Haley et al. succeeded in using exogenous TGF-β3 to reduce gastrointestinal side effects following radiation in mice [120,121]. In two studies, they demonstrated an increase in the number of viable intestinal crypts in animals receiving four 100 µg/kg doses of TGF-β3 intraperitoneal (i.p.) injections before 12–16 Gy abdominal irradiation. TGF-β3 treatment also decreased recovery time for damaged crypts. Protection was further increased by extending the TGF-β3 to two days instead of one. TGF-β3 treatment delivered orally did increase the number of surviving intestinal crypts after irradiation but was less effective than i.p. administration. After a dose of 15.8, but not 17.4 Gy, treatment with TGF-β3 increased animal survival from around 35% to around 95% after 30 days. The treatment also decreased the duration and severity of diarrhea in the 15.8 Gy group. TGF-β3 was shown through pulse-tritiated thymidine labeling to inhibit cell cycle progression of crypt stem cells, a finding that was corroborated by a decrease in intestinal crypt size.

Xu et al. demonstrated that weekly i.p. injections of 1 µg/kg TGF-β3 prevented radiation-induced pulmonary fibrosis in mice after a single 20 Gy thoracic irradiation [122]. At three and six months after irradiation, animals in the TGF-β3 treated group had lower fibrosis scores and decreased lung collagen content compared with untreated irradiated animals, although significantly higher than sham-irradiated. Histopathological indications of fibrosis appeared later and were less severe in the TGF-β3 treated group compared with non-treated irradiated animals. While irradiation alone caused an increase in the recruitment of fibrocytes to the lungs, TGF-β3 treatment partially counteracted this. Simultaneously, the percentage of Tregs in lung single-cell suspension was increased in the TGF-β3 treated group compared with irradiated and control groups.

In addition to the studies described in Table 3 and Table 4, several studies have indirectly observed the radioprotective effects of TGF-β3, by treating irradiated tissue with substances containing TGF-β3. In our laboratory, DBA/2 mice were LDR irradiated in a procedure shown to increase cytoplasmic TGF-β3 content in T-47D and T98G cells in vitro. When the animals were subjected to 9 or 9.5 Gy total body challenge irradiation six weeks later, the LDR-primed mice showed significant increases in survival [118,123]. Rong et al. demonstrated improved healing of radiation dermatitis in rat skin by subcutaneous injection of human fetal skin-derived stem cell (hFSSC) secretome compared with human umbilical cord mesenchymal stem cell (hUCMSC) secretome, and they observed increased TGF-β3 and decreased TGF-β1 mRNA expression in the hFSSC compared to the hUCMSC [124]. Finally, Borrelli et al. reduced skin stiffness and dermal thickness after irradiation of mouse skin by grafting fat enriched with CD74+ adipose-derived stromal cells, which showed increased expression of TGF-β3 [125]. 

### 4.3. The Role of TGF-β3 in Response to Low-Dose Irradiation

The low-dose region of ionizing radiation is associated with particular patterns of radiation response, one being the low-dose hyper-radiosensitivity (HRS) effect, which is presented as a relative hypersensitivity to radiation doses below approximately 0.5 Gy in about 80% of mammalian cell lines [126]. This effect can be transiently removed by submitting the cells to a so-called “priming” radiation dose of about 0.2–0.3 Gy at conventional high dose rates [127,128,129,130]. We discovered at our laboratory that delivering the same priming dose at a dose rate of 0.2–0.3 Gy/h permanently abolishes the HRS response to future challenge irradiations [131]. When primed at the low dose-rate (LDR) only, the irradiated cell conditioned medium (ICCM) from primed cells induces a transient removal of HRS in unprimed recipient cells in a protective bystander effect, indicating secretion of a radioprotective factor from the primed cells [132]. 

The permanent elimination of HRS in LDR-primed cells was found to be associated with increased content of TGF-β3 protein in the cytoplasm of T-47D and T98G cells [118]. In the same study, we found that neutralizing the antibody to TGF-β3 obstructed the removal of HRS by LDR-primed ICCM and that neutralizing TGF-β1 or TGF-β2 did not alter the effect. In corroboration with this, the addition of 0.001 ng/mL recombinant TGF-β3 to the medium of unprimed cells removed the HRS response to challenge irradiation when given 24 h before the irradiation. At a dose of 0.01 ng/mL, TGF-β3 treatment also induced protection to doses up to 5 Gy. 

In an in vivo study, removal of HRS was observed when serum from whole-body LDR-primed mice was transferred to unirradiated reporter cells, which displayed transient loss of HRS to subsequent challenge irradiation [133]. This mechanism was also blocked by adding a neutralizing antibody to TGF-β3 with the serum. 

We recently demonstrated that the permanent removal of HRS by TGF-β3 is mediated by binding to ALK1, independently of ALK5 and TGFβRII. We also showed that activation of TGF-β3 in this mechanism was dependent on an MMP family protein and that LDR-primed cells secrete TGF-β3 through EVs [21].

### 4.4. TGF-β3 and Chemotherapy

Cytotoxic chemotherapy is widely used to treat cancer, either alone or in combination with other treatment modalities, including radiotherapy. Although chemotherapy in contrast to radiotherapy acts systemically, the two treatments function similarly in their biological effects on the targeted tissue and cells. It is, therefore, reasonable to consider the effect of TGF-β3 in combination with cytotoxic drugs when reviewing its radiobiology. Table 5 summarizes relevant studies investigating the effect of TGF-β3 treatment in conjunction with chemotherapy.

Bleomycin is used both in chemotherapy and as an inducer of fibrosis in animal models. Khalil et al. observed increased expression of all three TGF-β isoforms after bleomycin-induced pulmonary fibrosis in Sprague-Dawley rats [134]. They also cultured explanted alveolar epithelial cells from these rats and found that these expressed TGF-β3 at a higher rate than TGF-β1 and -β2. When incubating the explanted alveolar epithelial cells with exogenous TGF-βs, all three isoforms inhibited proliferation.

Coker et al. treated mink lung epithelial cells with TGF-βs in vitro and found that although all isoforms increased procollagen production, TGF-β3 was the most potent [135]. However, when investigating mRNA expression after bleomycin treatment, no change in TGF-β3 expression was detected, and only TGF-β1 expression increased. 

Sonis, McCormack et al. investigated the effect of TGF-β3 on chemotherapy-induced oral mucositis in vitro and in vivo. In mink lung epithelial cell culture, pretreatment with 15 pM exogenous TGF-β3 for 24 h before the addition of one of a panel of chemotherapy drugs increased clonogenic survival of the cells. TGF-β3 treatment was effective against drugs that are cytotoxic in the S or M phase, but not against cisplatin or doxorubicin, which are cytotoxic throughout the cell cycle. TGF-β3 was shown to decrease cell proliferation in a manner depending on exposure time [136]. 

In another study by the same group, incubation with 50 ng/mL exogenous TGF-β3 for 24 h before the vinblastine challenge increased the proliferation of mink lung epithelial cells after 7 days [137]. In vivo, topical application of four doses à 20 µg TGF-β3 over 24 h reduced the proliferation of oral epithelial cells immediately after treatment. Proliferation recovered within 54 h. Submucosal injection of TGF-β3 had a similar effect on proliferation and reduced white blood cells count in a dose-dependent manner without affecting platelet number, an effect that was not seen after topical application. When given before 5-fluorouracil chemotherapy, TGF-β3 treatment improved oral mucositis score, reduced weight loss, and increased survival in the treated animals [137,138]. 

**Table 5 ijms-24-07614-t005:** Studies of treatment with exogenous TGF-β3 in combination with chemotherapy in vitro, in vivo and in clinical trials. HNSCC: Head and neck squamous cell carcinoma; 5-FU: 5-Fluorouracil.

Organ System	Model	Drug	Results	Reference
Lung	Sprague-Dawley Rat	Bleomycin	Endogenous TGF-β3 protein increased in rat lungs 28 days after bleomycin treatment. Incubation with exogenous TGF-β3 inhibited proliferation in explanted rat alveolar epithelial cells.	[134]
Lung	Human Fetal Lung Fibroblasts and B6D2F1 Mice	Bleomycin	Treatment with TGF-β3 increased procollagen production in cultured human fetal lung fibroblasts. TGF-β3 mRNA expression was stable after bleomycin-induced pulmonary fibrosis in mice.	[135]
Lung	CCL64 Cells	5-FU	Incubation with TGF-β3 for 24 h increased proliferation 7 days after cycle-selective chemotherapy.	[137]
Lung	CCL64 Cells	Vinblastine, Vincristine, Etoposide, Taxol, Ara-C, Methotrexate, and 5-FU	Pretreatment with TGF-β3 for 24 h increased cell survival in the clonogenic assay for chemotherapy drugs that were cytotoxic in the S- or M phase of the cell cycle.	[136]
Oral	HNSCC	Cisplatin and Paclitaxel	Knockout of TGF-β3 sensitized cells to chemotherapy. The addition of exogenous TGF-β3 to the knockout cells abrogated this sensitivity.	[20]
Oral	Syrian Golden Hamster	5-FU	Repeated topical application of TGF-β3 prior to chemotherapy reduced the severity of oral mucositis, reduced weight loss, and improved survival.	[137]
Oral	Syrian Golden Hamster	5-FU	Repeated topical application of TGF-β3 reduced the cycling of buccal epithelium and reduced white blood cell count after 3 h. Topical TGF-β3 treatment before chemotherapy reduced the severity of oral mucositis, reduced weight loss, and improved survival.	[138]
Oral	Human	Combinations of Cyclophosphamide, Epirubicin, 5-FU, Carboplatin and Thiotepa	Repeated treatments with TGF-β3 mouthwash before and during chemotherapy were well tolerated and neither systemic absorption nor the development of antibodies was observed.	[139]
Oral	Human	Any regimen with severe-grade oral mucositis incidence >50%	Repeated treatments with TGF-β3 mouthwash before and during chemotherapy did not influence the onset, duration, or severity of oral mucositis in 116 patients with breast cancers, lymphomas, and other solid cancers. There were no clinical differences between treatment and placebo regarding safety, and no evidence of systemic absorption of TGF-β3.	[140]

TGF-β3 containing mouthwash intended to reduce the severity of oral mucositis in cancer patients receiving chemotherapy reached phase II clinical trials but failed to demonstrate an advantage compared with placebo [139,140]. Patients’ assessments of oral pain, swallowing, or food intake were not improved by TGF-β3 treatment. However, repeated topical treatment with TGF-β3 at doses between 25 µg/mL and 100 µg/mL did not lead to systemic absorption or development of antibodies against TGF-β3, and no clinically relevant adverse effects of the treatment were noted. 

More recently, Rodrigues-Junior et al. showed that knockout of the TGF-β3 gene sensitized FaDu and SCC25 cells to the cytotoxic drugs cisplatin and paclitaxel [20]. The same effect was observed after inhibition of TGFβRII and reversed by the addition of exogenous TGF-β3. In the same study, they found that TGF-β3 protein levels in plasma EVs correlated with disease progression in patients with locally advanced head and neck squamous cell carcinoma. Crude plasma TGF-β3 did not correlate with disease progression. In vitro, cells resistant to cisplatin had significantly higher levels of TGF-β3 mRNA than cisplatin-sensitive cell lines. No such correlation was found for TGF-β1 or TGF-β2. EVs from cisplatin-resistant cell lines had a higher content of TGF-β3 protein compared with sensitive lines. Lastly, EVs from cisplatin-resistant cell lines decreased the drug sensitivity of sensitive lines when transferred, and this change in sensitivity was associated with increased Smad 2 phosphorylation.

## 5. Discussion

### 5.1. Variations in Endogenous TGF-β3 Expression after Irradiation

The role of TGF-β3 in radiation response is confounded by several factors: (1) in many studies, particularly early studies, all TGF-β isoforms are considered equal in function, despite a large body of evidence demonstrating otherwise. (2): TGF-β3 plays an important role in tissue homeostasis, and its expression may be a consequence of tissue injury rather than its cause. (3): Activation of the protein is often not evaluated in studies investigating radiation-induced changes in TGF-β3 expression. These factors are important to consider when interpreting results regarding TGF-β3 and radiation response.

Several studies have observed differing patterns of TGF-β3 mRNA and protein expression, emphasizing the importance of considering the difference between the two. O’Malley et al. observed a down-regulation of TGF-β3 mRNA after irradiation of rat mesangial cells but no corresponding change in TGF-β3 protein levels [105]. Yadav et al. observed similar increases in TGF-β3 mRNA and supernatant protein, but they detected a significant increase in TGF-β1 supernatant protein without a corresponding change in mRNA levels [108]. Discrepancies in mRNA expression and protein content may be a result of changes in secretion and ECM deposition or post-translational changes, including protein activation. Indeed, O’Malley et al. found that while TGF-β1 and TGF-β2 protein content increased after irradiation, the majority of the active TGF-β protein belonged to the TGF-β3 isoform. Investigation of downstream targets of TGF-β3 in addition to the protein itself, such as phosphorylation of Smad 2/3 or 1/5/8, may elucidate the true effect of radiation on the expression of active TGF-β3.

As evident from Table 1 and Table 2, a majority of the studies reviewed here report upregulation of endogenous TGF-β3 mRNA or protein after irradiation. However, several studies report downregulation. The conflicting evidence regarding the regulation of TGF-β3 after irradiation has several possible explanations. Differences in the expressing tissues, different intervals between irradiation and measurement, different radiation doses, and different model systems are all factors that are likely to influence the results. Notably, Ruifrok et al. observed disparities in TGF-β3 protein staining between intestinal crypts and villi in C3Hf/Kam mice: while an increase was observed after one day in the crypts, a decrease was measured in the villi after six days [112]. Several studies detected differences in TGF-β3 expression when repeating measurements at different time points. In rat liver, Seong et al. detected an increase in TGF-β3 mRNA one day after irradiation and a return to baseline after one week [115]. In mouse lung tissue, Finkelstein et al. observed an initial downregulation of TGF-β3 mRNA one day after irradiation, but an upregulation after two weeks [109]. Rubin, Johnston et al. observed down- and upregulation of TGF-β3 mRNA which appeared to be both dose- and time-dependent [110,111]. Lastly, Johnston et al. also observed a difference between mouse strains, where the radioresistant strain C3H/HeJ did not display any changes in TGF-β3 mRNA expression after irradiation. 

### 5.2. Radio- and Chemoprotective Effects of TGF-β3

The apparent role of TGF-β3 in radiation response has nominated it as an interesting target for modification of this response, mainly through exogenous addition in order to alleviate the side effects of radiotherapy. Several studies have indeed reported the radioprotective effects of TGF-β3 treatment (Table 3 and Table 4), and others have reported similar results after combined treatment with TGF-β3 and chemotherapy drugs (Table 5). Its radio- and chemoprotective effects have mainly been attributed to two different properties: its potential for cell cycle arrest and its anti-fibrotic action.

Given time before entering mitosis, mammalian cells possess a considerable capacity for DNA damage repair. By arresting the cell cycle of irradiated cells, TGF-β3 may provide that time, thus minimizing the radiation-induced damage to the cells. Several studies have observed an accumulation of TGF-β3 treated cells in the G1 phase, indicating inhibition of cell cycle progression at the G1/S checkpoint [88,119,134]. McCormack et al. demonstrated that TGF-β3 effectively protected cells from chemotherapy drugs that are cytotoxic in the S or M phase, but less so for drugs that are cytotoxic throughout the entirety of the cell cycle, supporting the claim that TGF-β3 increases cell survival through G1 arrest [136]. 

Potten et al. investigated different TGF-β3 treatment protocols in combination with irradiation of mouse intestine, varying the number of TGF-β3 injections and the time between each injection [120]. They observed that the addition of one TGF-β3 injection after irradiation, in addition to four injections before irradiation, decreased the crypt cell protection factor from 4.8 to 1.6. When TGF-β3 was administered at 4 time points from 8 to 32 h after irradiation, the treatment sensitized intestinal crypts to radiation damage, reducing the number of surviving crypts to around 1/3 of controls. They argued that these results support the hypothesis that TGF-β3 inhibits cell cycle progression in the intestinal crypt stem cells, allowing them time for DNA damage repair. However, when the inhibition was sustained after irradiation, the stem cells were not able to repopulate the crypts, thus increasing the radiation-induced damage. 

Ruifrok et al. argued that the decrease in TGF-β3 observed in intestinal villi after six days may have served the purpose of triggering proliferation after irradiation [112]. They proposed a regulation loop between nonproliferating villi cells and proliferating crypt cells: Radiation damage triggered a depletion in villi TGF-β3, suppressing the inherent crypt growth inhibition. As crypts started to regenerate, the crypt TGF-β3 production increased in order to establish growth inhibition of fully regenerated crypts. They corroborated this hypothesis with an observed inverse correlation between crypt cellularity and crypt TGF-β3 content.

The evidence regarding TGF-β3 and fibrosis is conflicting, assigning it both pro- and anti-fibrotic roles. The observed downregulation of TGF-β3 mRNA shortly after irradiation combined with upregulation of TGF-β1 observed in several studies introduced the idea that a balance between the two cytokines is inherent in tissue homeostasis and that a disruption of this balance may contribute to overexpression of ECM components and subsequent development of fibrosis [104,105,106,115]. Finkelstein, Rubin, Johnston et al. irradiated mice with doses in the range where development of pulmonary fibrosis is expected, and did indeed observe changes in TGF-β3 expression time points long before pathological changes can be observed, supporting the claim that such changes precede the development of fibrosis [109,110,111]. In corroboration with this, no changes were detected in C3H/HeJ mice, which are known to be resistant to the development of radiation-induced pulmonary fibrosis. However, they observed an upregulation of TGF-β3 mRNA, thus contradicting the hypothesis of an anti-fibrotic function of TGF-β3. Xu et al. successfully applied TGF-β3 treatment to reduce the severity of radiation-induced pulmonary fibrosis in mice [122]. They observed reduced fibrocyte recruitment, a shift in the interferon gamma/interleukin 4 (IL-4) balance in bronchoalveolar lavage fluid, and an increased number of Tregs in the lungs of TGF-β3 treated animals after irradiation. Together, these results point to a clear immunoregulatory and anti-fibrotic action of TGF-β3.

The possible importance of interplay between TGF-β3 and TGF-β1 is further complicated by the known diversity in downstream TGF-β pathways. Notably, both isoforms are known ligands of both ALK1 and ALK5, with phosphorylation of Smad 1/5/8 and Smad 2/3 as downstream effects. In a recently published study, we detected a competition between ALK1 and ALK5 for binding of TGF-β3, where ALK5 has the higher affinity, but TGF-β3 binding to ALK1 produces a radioprotective effect at low doses [21]. It is possible that the development of radiation fibrosis not only depends on the relative amount of TGF-β isoforms and their activation but also the relative expression of different TGF-β receptors. 

TGF-β3 dose and administration protocol are highly likely to affect both its radio-/chemoprotective and anti-fibrotic roles. Assuming a competition between receptors, a concentration that is high enough to saturate the higher affinity receptor may be necessary to produce the functional effects induced by binding to the lower affinity receptor. In the development of TGF-β3 mouthwashes to alleviate symptoms of chemotherapy-induced oral mucositis, the phase I study concluded with a dose recommendation of 100 µg/mL [139]. In the phase II trial, however, a dose of 25 µg/mL was used, and no protective effect was observed at this dose [140]. The authors in addition reported unpublished results from pig buccal epithelium, where 25 µg/mL TGF-β3 did not produce a protective effect. Together, these results suggest that the dose employed in the phase II trial was too low. 

### 5.3. TGF-β3 and Low Dose-Rate Irradiation—Sustained Induction or Perpetual Activation?

Low doses of ionizing radiation may induce specific effects that are not seen after higher doses, one of these being the HRS effect for doses below approximately 0.5 Gy. In a recent meta-analysis of 39 studies investigating gene expression after exposure to ionizing radiation, Sagkrioti et al. found that while doses in the 0.6–2.0 Gy range generally induced genes related to ROS metabolism, doses below 0.5 Gy were associated with the expression of genes related to cytokine and inflammatory response pathways [141]. 

Through a series of studies at our laboratory, LDR priming has been shown to permanently remove HRS from reporter cells in a TGF-β3-dependent mechanism [21,118,131,132,133,142,143,144]. The change in the amount of active TGF-β3 in LDR primed cells is thought to be small, as the effects resemble closely those produced by treatment with 0.001 ng/mL recombinant TGF-β3, but do not offer protection from radiation doses from 2 to 5 Gy, which is seen after treatment with 0.01 ng/mL TGF-β3. In addition, it was not possible to detect the increase in TGF-β3 by ELISA [118]. However, when serum from LDR-irradiated DBA/2 mice was transferred to reporter cells, a protective effect at higher radiation doses was observed, suggesting a more potent TGF-β3 upregulation or activation in vivo [133]. 

HRS is thought to be the result of a failure to induce the early G2 checkpoint after radiation doses below a certain threshold. This is supported by evidence of more prominent HRS in cells in the G2 phase [145,146] and a lack of HRS in G1 cells [131], and the fact that cells without inherent HRS response display activation of the early G2 checkpoint even for doses in the HRS range [145]. Removal of HRS by LDR priming in T-47D cells indeed caused a dose-dependent reduction in mitotic fraction at low doses, compared with a persistently high mitotic fraction for doses below 0.5 Gy in unprimed cells [142].

Several factors suggest that TGF-β3 is activated rather than upregulated after LDR priming. First, cell medium conditioned by unprimed cells and subsequently LDR primed was able to remove HRS from reporter cells, suggesting activation of a factor already present in the medium [132]. In addition, the amount of TGF-β3 protein was not different in EVs from unprimed compared with LDR-primed T-47D cells [21]. Finally, a broad-spectrum inhibitor of TGF-β activator proteinases MMPs removed the ability of an LDR-primed cell-conditioned medium to remove HRS in reporter cells [21].

In addition to TGF-β3, removal of HRS was found to depend on the action of IL-13 and its receptor IL-13Rα2, inducible nitric oxide synthase (iNOS), and peroxynitrite (ONOO-) [118,143,144]. IL-13 was suggested to be upstream of iNOS, as IL-13Rα2 has been shown to induce iNOS [147,148] and inhibition of IL-13 reduced the amount of total iNOS protein [144]. iNOS produces nitric oxide (NO), which again reacts rapidly with superoxide to produce ONOO-. NO has been observed to modify the activation of TGF-β1 by nitrosylation of LAP, thus inhibiting re-association between LAP and the active TGF-β1 [149]. Peroxynitrite has been shown to nitrate the aromatic amino acids tyrosine, tryptophan, and phenylalanine and may have a role as a scavenger for selective inhibition of LAP-3 in a way similar to that of NO for LAP-1 [150,151]. Interestingly, treatment with iNOS inhibitor 1400 W in an HRS-negative cell line was observed to induce HRS, suggesting a dependence on the iNOS/TGF-β3 pathway in inherently HRS-negative cells [143]. 

Finally, the removal of HRS by TGF-β3 was found to be a result of ALK1 binding, independently of ALK5 and TGFβRII [21]. When ALK5 was inhibited, however, HRS was removed without the addition of exogenous TGF-β3, indicating a competition between the two receptors, a higher affinity of ALK5 to the ligand, and an inherently low concentration of TGF-β3, which was sufficient to induce functional effects through ALK1 when the competition from ALK5 was removed.

## 6. Conclusions

Despite the high structural similarity with TGF-β1, TGF-β3 has different functions in many biological processes. However, TGF-β3 is much less studied than TGF-β1. TGF-β1 is activated by several integrins and members of the metzincin family, and some of these have been seen to also activate TGF-β3, but studies of specific activation of TGF-β3 by integrin and metzincin subtypes are wanting. TGF-β1, but not TGF-β2 and TGF-β3, was found to be activated by HDR irradiation [24]. However, TGF-β3 was activated by LDR irradiation in a mechanism dependent on iNOS activation [118,143].

A challenge when assessing the induction of TGF-β3 after irradiation is that measurements of mRNA levels, which is a common endpoint, or even latent TGF-β3 protein levels, do not necessarily correlate to activated TGF-β3 protein levels. Indeed, we showed that irradiated and unirradiated cells secreted the same amount of latent TGF-β3 protein in EVs, but it was only activated in the EVs from irradiated cells [21]. In addition, differences in dose and dose rate may influence the expression, secretion, activation, and duration of the activation of TGF-β3. In addition, different cells and tissues may induce or activate TGF-β3 at different time points after irradiation.

TGF-β3 has been found to affect radiation and cytotoxic chemotherapy toxicity both through cell cycle arrest and fibrosis mitigation. However, the effects appear to depend on TGF-β3 concentration in an interplay with TGF-β1 levels due to competition and different affinities for receptors ALK1 and ALK5. This complicates the potential use of TGF-β3 in protection against radiation- and chemotherapy-induced normal tissue damage and prompts more research into the combination of TGF-β3 with TGF-β1 and ALK5 inhibition.

Finally, there is a lack of clinical data regarding the expression and activation of TGF-β3 in response to ionizing radiation in humans and its potential radioprotective effects in human patients. Due to this, all available data originate from in vitro studies or animal models, and further studies are needed. 

## Figures and Tables

**Table 3 ijms-24-07614-t003:** Studies of treatment with exogenous TGF-β3 in combination with irradiation in vitro. HRS: hyper-radiosensitive.

Cell Line	Radiation Regime	Results	Reference
IEC6 and Widr Cells	0–10 Gy, ^137^Cs	TGF-β3 treatment resulted in decreased plating efficiency for IEC6 cells and increased for Widr cells. Increasing preincubation time with TGF-β3 reduced surviving fraction of Widr cells after 5 Gy. TGF-β3 treatment increased the radiosensitivity of Widr cells by reducing the shoulder region of the survival curve. TGF-β3 induced G1 arrest after 5 Gy in IEC6 cells and S-phase accumulation in Widr cells.	[119]
T98G and T-47D cells	0–5 Gy, ^60^Co	TGF-β3 treatment at a concentration of 0.001 ng/mL removed the default low-dose HRS of T-98G cells. At a higher concentration of 0.01 ng/mL, the treatment in addition increased radioresistance to doses up to 5 Gy for T98G and T-47D cells.	[118]

**Table 4 ijms-24-07614-t004:** Preclinical studies of treatment with exogenous TGF-β3 in combination with irradiation. CMC: Crypt microcolony assay.

Organ System	Model	Radiation Regime	Results	Reference
GI	BDF1 mice	Survival: 15.8–17.4 Gy, 300 kV X-rays; CMC: 8–16 Gy,^137^Cs	TGF-β3 injections before irradiation increased the number of surviving intestinal crypts for doses from 12–16 Gy. TGF-β3 injections increased survival after 15.8 Gy, but not 17.4 Gy abdominal irradiation.	[120]
GI	BDF1 mice	Survival: 15–17 Gy, 300 kV X-rays; CMC: 8–16 Gy,^137^Cs	TGF-β3 injections before irradiation increased survival and decreased recovery time of damaged intestinal crypts. Longer exposure to TGF-β3 increased the protection of intestinal crypts from radiation doses from 12–14 Gy. TGF-β3 treatment increased survival after 15.8 Gy abdominal irradiation.	[121]
Lung	C57BL/6 mice	20 Gy, ^60^Co	Weekly TGF-β3 injections after irradiation decelerated and decreased the severity of pulmonary fibrosis with a significant reduction in collagen deposition 3 and 6 months after irradiation. A significant reduction in the number of fibrocytes was observed 1 and 6 months after irradiation.	[122]

## Data Availability

No new data were created or analyzed in this study. Data sharing is not applicable to this article.

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
