# Peer review of "The Role of TGF-β3 in Radiation Response"

_ijms, 2023, doi:10.3390/ijms24087614_

Round 1
Reviewer 1 Report
This is a nice review study on the Role of TGF-beta3 an isoforms of the protein. It is not the first time that TGF-beta in Radiation response has been investigated but mainly the beta1 isoforms.
The study is a comprehensive nice review lacking though of some points.
1. Some years ago a representative cumulative bioinformatics study for the association between radiation response with inflammatory and immune system response in various human tissues malignant and non
Please see the work :Antioxidants (Basel) 2022 Nov 18;11(11):2286. doi: 10.3390/antiox11112286.
In addition, there is not direct evidence for the existence of the protein at low or high levels in human organisms, radioprotective or Radiosensitizing Effects and overall mechanistic etiology for the role in humans. The authors are asked to comment and revise accordingly.
Last but not least, what is the prevalence of each isoform in humans and patients more specifically?
Reviewer 2 Report
This review paper is about the role of TGF-β3, focusing on the radiation response. The cell cycle-regulating and anti-fibrotic effects of TGF-β3 have been identified as a potential mitigator of radiation- and chemotherapy-induced toxicity in healthy tissue.
This is a comprehensive review by citing various publications about TGF-β3, including conflicting reports. TGF-β3 is much less studied than TGF-β1, and its function is not well understood. It has potential radioprotective and anti-fibrotic effects and might be a useful for cancer therapy or radiation protection.
However, this manuscript is not well-organized possibly due to the high volume. Important points should be summarized in a more comprehensible manner, taking into account what can be written briefly and what can be omitted.
Round 2
Reviewer 2 Report
This manuscript can be accepted.